# Implementation of Check-In/Check-Out to Improve Classroom Behavior of At-Risk Elementary School Students

**DOI:** 10.3390/bs13030257

**Published:** 2023-03-14

**Authors:** Ashley L. Sottilare, Kwang-Sun Cho Blair

**Affiliations:** 1Santiago & Friends Family Center for Autism, Orlando, FL 32807, USA; 2Applied Behavior Analysis Program, Department of Child and Family Studies, University of South Florida, Tampa, FL 33612, USA

**Keywords:** check-in/check-out, tier 2, daily report card, MTSS, PBIS

## Abstract

The multi-tiered school-wide positive behavioral interventions and supports offers a comprehensive model for the prevention of behavioral and academic problems in schools. This study evaluated Check-in/Check-out (CICO), a Tier 2 intervention, with three elementary school students from a high-need population, whose problem behavior was hypothesized to be maintained by teacher attention. The study employed a concurrent multiple baseline design across participants, a single case experimental design to examine the effects of CICO on student academic engagement and problem behavior during instruction. Results indicated that implementation of CICO with fidelity by the CICO coordinator, classroom teachers, and parents lead to increased academic engagement and reduced problem behavior in all three students. Tau-*U* Effect sizes were medium to large for academic engagement and small to medium for problem behavior across students. Data on two students indicate that systematically fading the number of times teachers utilize the daily report card has the potential for promoting maintenance effects. For one student, fading of the coordinator was successful. Social validity assessment indicated high levels of satisfaction with the CICO intervention by the participating students, teachers, and one parent.

## 1. Introduction

The multi-tiered school-wide positive behavioral interventions and supports (SWPBIS) utilizes three tiers of prevention and intervention, which has been shown to have a positive impact on student behavior and academic performance at all three universal, secondary, and tertiary tiers [1,2,3,4]. The primary level (Tier 1), also referred to as universal supports, focuses on the prevention of problem behaviors through the development of school-wide expectations that encourage appropriate behavior. Out of the student population, typically 80% are likely to respond to Tier 1. The secondary level (Tier 2) includes targeted interventions for students (approximately 10% to 15% of students) who need supplemental supports due to the first tier of intervention being ineffective on its own. The remaining ~5% of students require support at the tertiary level (Tier 3), which involves providing intensive individualized interventions for their severe problem behaviors [3,4]. The SWPBIS framework provides guidance for schools to select and implement evidence-based practices; however, effective and sustained implementation of the SWPBIS requires schools to utilize data systems to determine the level and type of intervention needed for individual students [5]. Of the three tiers, Tier 2 interventions are imperative for students who have emerging social–emotional needs and whose behavior requires an additional level of supports.

Three elements have been identified as being crucial to effective Tier 2 interventions: (a) team-based system of planning, (b) intervention practices that are available and known by the school personnel, and (c) use of data to make program decisions [3]. Tier 2 interventions focus on practices that: (a) are highly efficient and delivered similarly across groups of students who are likely to benefit from the same type of intervention, (b) increase structure of measuring outcomes in the form of observable events to facilitate efficient monitoring, and (c) increase intensity of instruction on expectations and desired behavior [6]. The primary goal of Tier 2 interventions is to address the needs of students who are at-risk for developing severe problem behavior and prevent their future problems [1]. Since Tier 2 interventions are efficient and require a low level of effort to implement, schools can better allocate resources in comparison to more time intensive Tier 3 interventions [7]. The common evidence-based Tier 2 behavior interventions that have been used in schools include Check-In/Check-Out (CICO) [8,9,10], social skills training [11,12], self-monitoring [13,14], and group contingencies [15,16].

Of the Tier 2 interventions, the most frequently used Tier 2 intervention in U.S. schools is CICO, also referred to as the Behavior Education Program [8,9,10]. Benefits of this approach include: (a) increasing antecedent prompts for appropriate behavior, (b) increasing adult feedback, (c) providing a more structured daily schedule for students, and (d) improving the school feedback to families with respect to the students’ behavior goals [17,18,19]. The CICO intervention has the following main components: (a) students “check-in” with an adult (the CICO coordinator) in the morning, receive their daily report card (DRC; also referred to as a daily point card), and review the day’s expectations; (b) students receive ongoing feedback from teachers and staff throughout the day; (c) students “check-out” in the afternoon and review their DRC with the CICO coordinator, discussing the level of success; and (d) the DRC, or a summary of the DRC, is sent home for parents to sign and return the following day. Delivery of reinforcement for meeting daily goals is also included [20,21]. When the student checks-out and discusses their overall behavior performance, the coordinator assesses the student’s progress by quantifying the student’s daily report card scores. A pre-established goal is put in place by the CICO team in advance to determine whether the day was a success overall [22].

Recent reviews on CICO indicate that CICO has sufficient evidence to be regarded as an evidence-based practice within SWPBIS [20,23,24]. The CICO intervention has been found to be effective for improving classroom behavior for students who are at-risk for developing severe problem behaviors, particularly for those whose function of problem behavior is adult attention [20]. Many of the studies on CICO included social validity assessments and showed that, overall, the CICO intervention was well received by teachers and parents, e.g., [9,18,25]. However, of the current literature on CICO, the treatment fidelity reported are often limited and do not delve into the quality of implementation, but instead focus only on the overall implementation [9,19,22,25,26,27]. The components most frequently assessed for accurate implementation are: (a) student checks-in at the start of the day; (b) DRC is marked by the teacher; (c) student checks-out at the end of day; (d) parent initialed DRC; and (e) data are assessed by the implementation team [17,20]. Although these are key components, few studies have assessed whether the teachers are attending to all of their implementation components (i.e., mark DRC at the designated intervals throughout the day, give student verbal feedback), and whether the assigned school personnel at check-in/out are implementing all required steps (i.e., check to see whether student has materials, review expectations for the day with student, use positive statements, and provide reinforcers for points at check-out). These are all essential components of the CICO intervention [22] and should likewise be assessed for accurate implementation.

Furthermore, studies seldom assess family review and feedback [19,20,25]. Often the fidelity or treatment integrity measure did not include the parent review component. Filter and colleagues [19] reported that of 17 students from 3 elementary schools, whose fidelity of implementing the CICO intervention was assessed, only 3 students’ data (17.6%) showed that a family member reliably initialed the DRCs. Additionally, only 41% of respondents, consisting of teachers, administrators, and staff, reported that the family feedback component was utilized. Hawken and colleagues [25] reported that family review and feedback only occurred 36% of the time in their study. Similarly, in a systematic review of group design and single case design studies on CICO, Maggin and colleagues [20] found that of the eight single case design studies that met the What Works Clearinghouse evidence standards with reservations, only two studies (25%) reported fidelity assessment of the family involvement (home–school collaboration) component.

Considering that the schools operating with a high degree of home–school collaboration and those that utilized the DRC more often throughout the day demonstrated stronger student outcomes than those that did not [28], procedures for increasing the parent use of the DRC should be sought out to enhance the outcomes of the CICO intervention for students who are not showing adequate desired behavior change, particularly when the parent involvement is not consistently being utilized. In recent years, DRC has been used not only as a student progress monitoring tool, but also as a behavioral intervention in schools. Reviews of studies on DRC indicate that using DRC as interventions can improve student behavior significantly [29,30,31]. When implementing CICO, the DRC serves as a visual prompt for the student throughout the day and promotes family involvement in monitoring their child’s academic and behavioral progress. Based on the child’s success throughout the day, parents can choose to carry the day’s behavior consequences to the home environment, which will contribute to enhancing their child’s academic and behavioral outcomes.

Another limitation of the current CICO literature is the limited research demonstrating the intervention maintenance effects. In a component analysis of CICO with four elementary school students, Campbell and Anderson [9] systematically faded out the intervention after maintaining a high percentage of possible DRC points by decreasing the intervals, at which the students received teacher feedback throughout the day (two, one, and zero). One student made it to the zero level where only the CICO components were in place without teacher feedback, whereas three students required a return to an increased level of teacher feedback to maintain low levels of in-classroom problem behaviors. While the authors speculated that the most powerful components of CICO might have been the morning check-in and afternoon check-out by the coordinator, fading of these components should also be assessed. Once the DRC has successfully been faded out, the classroom teachers could incorporate the use of DRC into their classroom management techniques already in place.

Given the scarce literature available assessing implementation fidelity for all CICO components and evaluating maintenance effects, the present study attempted to extend the CICO literature by facilitating all aspects of the CICO implementation fidelity and evaluating the maintenance effect targeting students at-risk for emotional and behavior disorders served in a high-need elementary school setting. The study addressed the following questions: (a) to what extent can the CICO intervention be implemented with fidelity by all team members (i.e., teacher, coordinator, and parent); (b) to what extent will the implementation of CICO increase academic engagement and decrease problem behavior in elementary school students at-risk for developing severe problem behavior; and (c) to what extent can the CICO intervention be faded out while maintaining positive student outcomes?

## 2. Materials and Method

### 2.1. Setting and Participants

This study was conducted at a high-need public elementary school located in a low-income residential area of a city in Florida. The school’s population was approximately 420 students and was listed as a Title 1 school with more than 75% of the students receiving free or reduced-price lunch. Of the student population, approximately 66% were Black, 24% were Hispanic, and 9% were White. The school consisted of three to five classes in each grade level with an average of 17 children per class. The school received a 95 out of 107 points on the Benchmarks of Quality-Revised [32], which signifies that the universal Tier 1 supports within SWPBIS were being implemented with fidelity. Even though the school had been implementing the Tier 1 supports with fidelity for more than 3 years, the school still had behavioral concerns. The school’s behavior support team reported that in the previous academic year, before the study began, the school had a total of 93 major office discipline referrals (ODRs).

The primary participants in this study consisted of three first-grade male students and their two classroom teachers. The secondary participants were their three parents and the school CICO coordinator (guidance counselor) and a second guidance counselor who filled in on few sessions when the original CICO coordinator was out. All student participants were from single-parent African-American families with incomes below the poverty line and were receiving free lunch at the school. Through implementation of SWPBIS, the school’s PBIS team identified the students who were not adequately progressing while receiving Tier 1 supports, having moderate problem behavior. Selection criteria for students included: (a) having received a major ODR or 2 to 5 minor ODRs in the current academic school year; (b) exhibiting problem behavior that occurred throughout the day; and (c) the potential function for problem behavior was hypothesized to be attention. Students were excluded from the study if their problem behavior was dangerous to themselves or others and did not occur in multiple locations or time periods, and if adult attention was found to be aversive. Teachers were eligible for participation if they had at least one, but no more than two students, who would benefit from the CICO intervention. Once students were deemed eligible, a meeting was held to explain the study and attain parental and teacher informed consent and student assent. Students were screened using the Functional Assessment Checklist for Teachers and Staff (FACTS-A & B) [33] by their teachers to ascertain whether their problem behaviors were in fact in line with the selection criteria. The results of FACTS were corroborated with results of two 30 to 40 min observations conducted by the first author during the academic time period when problem behaviors occurred most often.

#### 2.1.1. Students

Evan was 7 years old and did not have any known developmental or medical diagnoses. On the statewide reading assessment, he was performing in the 75th percentile in relation to his peers. It was reported that Evan engaged in frequent out-of-seat behavior, verbal classroom disruption (e.g., crying, whining, and grunting), and nonverbal classroom disruption (e.g., throwing items and kicking or hitting furniture or objects) often requiring the school’s behavior specialist to address problem behaviors with the teacher three to four times per week. Although these behaviors occurred throughout the day, the teacher had indicated in the FACTS that they most frequently occurred in the morning during writing. This was corroborated by direct observations by the first author. During observations, it was also observed that Evan’s problem behaviors occurred at higher rates when the teacher did not provide attention immediately following Evan’s request or hand raise. Peer attention was also observed to be reinforcing. During situations in which Evan was given the opportunity to be in front of the class for a special task, garnering attention, his problem behaviors occurred less often.

Jeremy was 6 years old and did not have any known developmental or medical diagnoses. On the statewide reading assessment, he was performing at the 10th percentile in relation to his peers. The school’s PBIS team referred him for problem behavior that occurred throughout the day. During observations, Jeremy frequently engaged in off-task behavior, not attending to the teacher or materials, and engaging in inappropriate sitting (e.g., lying across the chair, sitting or laying on the floor, and rocking the chair) and physical classroom disruption in the form of hitting and throwing items, sweeping items off the desk, and removing items from his desk cubby or backpack onto the floor. The teacher required the behavior specialist’s support in the classroom to address Jeremy’s problem behaviors on a weekly basis. Based on the FACTS and direct observations by the researcher, it was determined that Jeremy’s problem behavior was most likely to occur during class instruction and activities that took place in the morning until lunch, with reading time being most problematic.

James was a typically developing 7-year-old student from the same class as Evan. He also did not have any known developmental or medical diagnoses. On the statewide reading assessment, James was performing in the 10th percentile in comparison with his peers. James was referred for problem behavior that was observed to be in the form of physical classroom disruption (e.g., hitting, kicking, throwing items, and knocking over furniture), frequent out-of-seat behavior, lying on the floor, and was often off-task, not attending to the materials or the teacher during times of instruction. The behavior specialist was being utilized to address James’ problem behaviors on a weekly basis. Information obtained through the FACTS and observations indicated that unstructured activities were especially problematic for James. During center activities in which students were permitted to work with peers in designated locations around the room, James was most frequently observed engaging in problem behavior, which resulted in receiving peer and teacher attention.

#### 2.1.2. Teachers

Two first grade teachers participated in the study. One student (Jeremy) was from Teacher 1′s first grade class consisting of 18 students (72% African-American, 17% Hispanic, and 11% Caucasian). Teacher 1 was a novice female teacher with 3 years of teaching experience. This was her 2nd year teaching at a school implementing SWPBIS. Her classroom management strategies included: (a) posting the SWPBIS expectations and rules; (b) teaching and referring to the expectations and rules throughout the day; (c) assigned seating that was easy to maneuver while not providing access opportunities for elopement; (d) using a classroom level system in which students’ name cards are moved up a level at the end of the day for good behavior, working towards a classroom party or activity once the whole class reached a set goal; and (e) using a token economy, which was integrated into the overall schoolwide Tier 1 token system.

Two students (Evan and James) were from Teacher 2′s first grade class consisting of 17 students (82% African-American, 12% Hispanic, and 6% Middle Eastern). Teacher 2 was also a novice female teacher with 2 years of teaching experience. Her classroom management strategies included: (a) posting the SWPBIS expectations and rules; (b) teaching and referring to the expectations and rules throughout the day; (c) posting a daily schedule that is discussed with the students at the start of each day; (d) using a five-level color system in which students’ clips were moved up and down throughout the day in response to their behavior with students receiving tokens at the end of the day for finishing on the third level or higher; and (e) using a token economy that was integrated into the schoolwide Tier 1 token system. Teacher 2 was also observed giving frequent praise for desired behaviors and utilizing pivot praise. During these times she would turn to a student who was engaging in the desired behavior and publicly acknowledge or praise them while paying little to no attention to students engaging in undesired behaviors, quickly turning back to provide them praise once they began to engage in the desired behavior.

#### 2.1.3. Parents and CICO Coordinator

Evan’s mother was a single parent in her mid 30s. Teacher 2 reported that communications were difficult and that papers and materials sent home requiring parental attention or signature were often not reviewed or returned unless a phone call was made to the parent. Jeremy’s mother was a single parent in her late 20s. Teacher 1 reported that communication with home were successful approximately half of the time. James’ mother was a single parent in her mid 40s. Teacher 2 reported that James’ mother was often disinterested in collaborating with the school on addressing James’ problem behavior, frequently declining to meet on campus with the teacher and administrators. Home–school communication through notes in the student’s planner and school flyers were successful about half of the time in eliciting a response. The CICO coordinator was a school guidance counselor who was female in her early 40s. She had been working at the school for 3 years and served on the school’s PBIS team. The second CICO coordinator was also a school guidance counselor who was female in her mid 50s. She had been working at the school for 2 years.

### 2.2. Measurements

#### 2.2.1. Academic Engagement and Problem Behavior

Student problem behavior and behaviors linked to academic engagement were both identified and operationally defined, based on the results obtained through the FACTS questionnaire and two 30 min direct observation sessions conducted prior to baseline data collection. All target behaviors were recorded during 30 min sessions using a 15 s partial interval recording procedure. The percentages of intervals for academic engagement and problem behavior were measured. For Evan, academic engagement was defined as sitting appropriately (e.g., sitting on bottom in assigned desk chair or sitting on bottom or knees in assigned spot on floor for longer than 8 s) and problem behavior was defined as verbal or nonverbal classroom disruption (e.g., verbal echoing of teacher’s dialogue without explicit instruction from the teacher to do so, crying, vocalization above the classroom’s volume or attempting to engage peers in conversation during instruction or independent work, throwing items at others, desk, or floor, hitting one item with hand or other item, and kicking or knocking over items or furniture).

For both Jeremy and James, academic engagement was attending defined by eyes are directed toward the teacher or instructional materials presented by the teacher, interacting with assigned task materials, or working on assignments or completing independent work. Problem behavior included inappropriate sitting and disruption. Inappropriate sitting was defined as putting feet on the desktop, chair, or counter, kneeling in seat, rocking or leaning the chair, laying across the chair or desk on back or belly, lying on the floor on back or belly, or failing to remain in the assigned seat, being out of the assigned area. Classroom disruption was defined as attempting to throw items at others, desk, or floor, throwing items, hitting one item with hand or item, or kicking or knocking over items or furniture.

Direct observations occurred 3–5 days per week for each student and took place during the academic time period when problem behavior occurred most often. Observations were conducted using a smartphone application called Interval Timer to signal the intervals within 30 min sessions. Intervals were set to occur every 15 s throughout the 30 min observation sessions, at which time the timer would buzz lightly or make a quiet tick sound. If the observer was unable to easily hear the signal sound, earphones were permitted so that increasing the volume would not disturb the classroom being observed.

#### 2.2.2. Implementation Fidelity

To assess correct implementation of CICO procedures, three areas of fidelity were assessed: teacher implementation, CICO coordinator implementation, and parent use of DRC. Teacher and CICO coordinator implementation fidelities were assessed for 24% of intervention sessions using yes/no checklists by directly observing their sessions by the researcher. Teacher fidelity checklist included 5 items: (1) completing the DRC after each assigned class period; (2) treating each class period independently; (3) reviewing what was inappropriate and how to behave better if a score of “0” was received; (4) giving student praise and reviewing how to do better next time if the score of “1” is received; and (5) giving student praise if a score of “2” is received. The school guidance counselor, who served as the CICO coordinator, was assessed for treatment fidelity on 11 items: (1) asking or prompting student to show the DRC at check-in; (2) checking for parent signature and checklist; (3) providing praise for turning in a signed card or reminding student for the next day; (4) asking student to identify the expectations; (5) reviewing student’s point goal for the day; (6) collecting the DRC from student at afternoon check-out; (7) reviewing the day’s progress; (8) tallying DRC points; (9) asking student to identify the expectations again; (10) reminding student to have parents sign the card; and (11) concluding the session with a positive statement. Further implementation fidelity was assessed by measuring parent use of DRC for 100% of intervention sessions, which focused on whether the parents reviewed the DRC with their child as indicated by parent signature on the returned card and provided feedback to the coordinator via parent response field on the returned card. The parent fidelity assessment was conducted indirectly in each session when student arrived at check-in by recording whether the student returned the previous day’s DRC with parent signature that noted whether their child remembered to give them the DRC or they had to remind the child to give them the DRC. For each child, the percentage of steps implemented correctly by teacher and coordinator and the occurrence of the parent providing a signed DRC were measured to assess the implementation fidelity for each CICO component.

#### 2.2.3. Social Validity

At the conclusion of the study, participating students, parents, and teachers were given the student, parent, and teacher versions of the Behavior Education Program Acceptability Questionnaire [8] to assess social validity, which used a Likert-type scale (1–6) with higher scores indicating higher acceptability of the intervention procedures and satisfaction with the intervention outcomes. For students, 8 items were assessed, including if they felt it helped change their behavior, if they would want to participate again, and if they were likely to recommend the intervention to other students. For parents, 7 items were assessed including whether they felt the intervention helped keep them informed of their child’s behavior, if they felt the intervention was successful in increasing academic performance, and if they felt their child’s behavior had improved. For teachers, 7 items were assessed with an additional option to write in comments at the end. Questions addressed whether the CICO was worth the time and effort, whether they felt there was a significant change in behavior, the ease of implementation, and if they would recommend the program to others.

#### 2.2.4. Interobserver Agreement

To assess interobserver agreement (IOA), two observers (an undergraduate student in psychology and a graduate student in an applied behavior analysis) independently and simultaneously recorded data for 30% of all direct observation sessions across participants, behaviors, and experimental conditions. The observers were trained in data collection procedures using an instructional training video created by the researcher (first author), which included a practice observation test, until 90% agreement was achieved during a 10 min observation test period. Percentage of agreements was calculated by dividing the number of agreements by the number of agreements plus disagreements multiplied by 100. Mean IOAs for baseline, intervention, and fading across student participants were 97.4%, 96.3%, and 97.6%, respectively, ranging from 93.8 to 98.3%. Mean IOAs for appropriate behavior and problem behavior across students were 96.2% and 97.3%, respectively, ranging from 95.3 to 98%. The IOA for implementation fidelity was 100% across coordinator, teachers, and parents.

### 2.3. Experimental Design and Procedures

The outcome of the CICO intervention was tested using a concurrent multiple baseline design across participants. Intervention was systematically staggered across students collecting a minimum of 7 baseline data points. We planned to introduce intervention first to the student with baseline data showing the most stability or showing a counter therapeutic trend for both academic engagement and problem behavior. Jeremy showed the most stability after 5 baseline sessions; however, intervention was introduced to Evan first due to his urgent need for intervention. A preference assessment was conducted using a 42-item student survey with a 0–2 rating scale to ensure that the items available for purchase with their CICO points were desired and likely to act as reinforcers. The researcher and each classroom teacher jointly developed DRC. The DRC’s format consisted of a grid in which each teacher’s class schedule was broken down into six main subjects and listed in a row along the top. The school’s five READY expectations (*Respectful, Eager to learn, Active learner, Directions I follow them*, and *Yes to safe choices*) were listed in a column along the left side. An example on what to do was written next to each expectation (e.g., nice hands and feet and walking feet), which met the teachers’ classroom needs.

#### 2.3.1. Teacher and Staff Training

The researcher provided 20 min training to the participating teachers, CICO coordinator, and the school’s PBS team members on the basics of the CICO intervention and the referral process. The training consisted of a PowerPoint presentation and elements from the DVD of the CICO manual [34] to gain school-wide support for the CICO intervention. Once students had been selected for participation, their teachers received additional one-on-one training (20–30 min session) prior to baseline data collection, reviewing the CICO intervention procedures, their role in the implementation process, and how to use the DRC point card.

#### 2.3.2. Baseline

During baseline, all student participants continued to participate in the school’s universal supports as part of SWPBIS. The students were instructed on the school’s expectations in the classroom. Additionally, the school utilized a school-wide token economy in which the students could earn paper tokens throughout the day for engaging in appropriate behaviors. All teachers and other school personnel participated in administering the tokens to students throughout the day, and each class had tangible reinforcers for which students could purchase with their tokens on designated days. Students also had the option of shopping at the school store or buying special treats (e.g., ice pops, popcorn, and sweets) on a monthly basis. Students continued to receive these universal supports and discipline measures throughout the study. Baseline data were collected 3–5 times per week for a period of approximately 2 weeks during the most problematic activity period of the school day. The baseline data were collected across 11 days for Ethan, 12 days for Jeremy, and 14 days for James, excluding weekends. At the conclusion of baseline data collection, the researcher met with each teacher to briefly review the CICO process and their role. Both teachers easily recalled their responsibilities and did not express any additional concerns.

#### 2.3.3. Implementation of CICO

The CICO intervention consisted of morning and afternoon meetings with the coordinator and regular feedback from teachers using a DRC at designated intervals throughout the day. The DRC point goal for each student, initially determined based on baseline performance, was used to assess the students’ progress each day. The goal was set for 70–85% of the daily possible points depending on the students. The intervention was implemented across 44 days for Ethan, 30 days for Jeremy, and 42 days for James, excluding weekends. At morning check-in, each student arrived at the designated CICO meeting room and met with the CICO coordinator. At this time, the student returned the previous days’ DRC with parent signature to the coordinator. The coordinator then checked to see if the student had the materials (pencil, paper, and planner). The coordinator provided verbal positive feedback for remembering to come to check-in, bringing the signed DRC, and having their materials. If the DRC was not returned signed by parent, the coordinator talked to the student about remembering for the next day. The coordinator and student then reviewed the school’s expectations and discussed the day’s point goal. The student was given a DRC to be delivered to their classroom teacher. Once morning check-in was completed, the coordinator sent the student off to class. Teachers scored the students after each designated class segment using a 0–2 scale for each behavioral goal on the DRC point card.

At the end of the day, the student returned to the CICO location and checked-out with the coordinator. At this time, the student provided the coordinator with their completed DRC for review. The coordinator and student discussed the day’s accomplishments and areas requiring additional efforts. The coordinator tallied the student’s DRC points and rewarded the student with the correlating incentive for the attained score. The coordinator provided parents with a letter explaining their role in implementing the intervention including how to provide feedback to their child. The DRC included an additional place for parent response and a place for parental signature. Parent implementation fidelity was ensured by reminding the students to have a parent sign the DRC card and providing verbal praise when this was achieved. Prior to starting the intervention, each student attended a 20-min training session conducted by the researcher. The sessions consisted of: (a) instruction regarding what to do with the DRC card at school; (b) modeling of the routine by the researcher; (c) a chance for students to rehearse the skills; and (d) feedback from the researcher.

The researcher observed 24% of the CICO implementation sessions in the CICO meeting room and in the classroom to assess implementation fidelity and provide the coordinator and teachers with verbal performance feedback at the end of the session. Because the implementation fidelity was 100% in all observation sessions, additional training on intervention procedures was not provided to the coach and teachers. It was planned that a booster training session would be conducted implementing modeling, rehearsal, and feedback components of behavioral skills training (BST; [35]) until they demonstrate 100% accuracy if the fidelity fell below 80%. The fidelity goals for the coordinator and teachers were three consecutive sessions with at least 80% correct implementation of the procedural steps.

Out of the three participating students, Evan, was shown to be unsuccessful with the standard CICO intervention, showing minimal changes to target behaviors. This was thought to be due in part to failing to turn in the previous day’s DRC or showing a lack of parental use of the DRC and failing to bring materials to school (e.g., paper, backpack, and planner). To address these concerns, the intervention was modified adding an additional accountability tracking system to the standard CICO. The additional procedure involved using a 5-item accountability tracking sheet during the morning check-in routine, which assessed whether Evan: (a) brought the card to morning check-in; (b) had the card signed by parent with a check mark indicating that Evan showed parents the card without a reminder; (c) brought school materials with him; and (d) brought the signed DRC throughout the week.

Evan received an additional 20 min training prior to the implementation of the accountability tracking. This training session consisted of: (a) instruction regarding what to do with the new DRC and accountability sheet at school and at home (e.g., being responsible for giving card to their parent and having them sign); (b) modeling of the routine by the researcher; (c) a chance for Evan to rehearse the skills; and (d) feedback from the researcher. The coordinator reviewed with him the checklist items on the accountability tracking sheet and explained his responsibilities for having his parents review and sign the DRC. The accountability tracking provided an additional chance to come into direct contact with reinforcement at the start of the day.

At the beginning of the morning check-in routine, the coordinator reviewed the accountability tracking sheet with him. Evan had the opportunity to earn additional points that were exchanged for additional incentives if he brought the DRC with parent signature and school materials. The incentives were chosen based on the preference assessment conducted in baseline and were available for purchase with the additional incentive points at the morning check-in. During the afternoon check-out, Evan was given a copy of the accountability tracking sheet to take home to provide a visual reminder of his responsibilities.

#### 2.3.4. Fading

For two students (Evan and James) who met their set point goals for at least 5 consecutive days, a fading procedure was implemented. The fading procedure was implemented gradually by systematically decreasing the number of times teachers utilized the DRC in the classroom. For example, if the teacher originally scored the student at 5 different time periods throughout the day, they would then move down to 4 feedback periods, and then 3. The final step in the fading procedure involved removing the CICO coordinator from the daily routine during which the teacher continued using the feedback period and parent-home note without the coordinator’s morning check in and afternoon check out. This final fading procedure was implemented with Evan. James showed an undesired behavior change during the second fading phase and was returned to first successful fading phase for additional time until he had consistently returned to a desired level of academic engagement and problem behavior. The fading procedures were implemented for 12 days for Ethan and 13 days for James.

### 2.4. Data Analysis

In addition to visual analyses of data, which examined level, trend, and variability of data within and across phases, immediacy of effects, consistency of data, and staggered treatment effects across tiers, Tau-*U* effect sizes [36] were computed using a web-based calculator (https://jepusto.shinyapps.io/SCD-effect-sizes/; accessed on 2 February 2023). Although a Tau-*U* effect size of 1.0 indicates nonoverlap of all pairwise comparisons, its values may be inflated [37]; therefore, Tau-*U* effect size estimates should be interpreted with caution. In interpreting the Tau-*U* indices, we used the following guidelines: <0.20, small effect; 0.20–0.59, moderate effect, 0.60–0.80, large effect; and >0.80, very large effect [38].

## 3. Results

### 3.1. Implementation Fidelity

Scores on the implementation fidelity checklists indicated that both the coordinator and teachers implemented the CICO procedures correctly as designed throughout the intervention phase. Additionally, a school guidance counselor who served as an additional trained CICO support coordinator on the days the primary coordinator was unavailable demonstrated correct implementation of the all required components, suggesting high coordinator implementation fidelity. The implementation fidelity was 100% for coordinators and teachers in all sessions. For parent implementation fidelity which was assessed indirectly though reviewing DRCs submitted by students showed that in both intervention and fading phases, the parents’ use of DRC as designed averaged 80% of the sessions, ranging from 75–83% of the sessions for Jeremy and James. The parent use of DRC was 13% of the sessions for Evan during the initial implementation of CICO and increased to 92% when the five-item accountability tracking sheet was used during the morning check-in routine.

### 3.2. Student Behaviors

Figure 1 displays data on student behaviors. Implementation of the CICO resulted in an immediate increase in academic engagement and immediate reduction in problem behavior across students. For Evan, a relatively high level of academic engagement and low level of problem behavior were observed during the initial baseline sessions; however, by day 5 both target behaviors returned to the level reported by the teacher during the FACTS interview, averaging 20% (range = 13–29%) of intervals with academic engagement and 39% (range = 33–53%) of intervals with problem behavior during the last three baseline sessions. Data showed a decreasing trend for academic engagement and an increasing trend for problem behavior. Overall baseline mean level was 47% (range = 13–87%) for academic engagement and 23% (range = 4–53%) for problem behavior. It is likely that the higher level of academic engagement and lower level of problem behavior during the initial phase of baseline than those reported by the teacher were the result of reactivity to the presence of the observers.

Implementing the CICO resulted in an immediate increase in academic engagement with an average of 62% (range = 37–87%). Problem behavior decreased immediately following the implementation of CICO; however, it continued to occur during an average of 22% (range = 6–47%) of intervals with a high variability. Once Evan received additional supports through the accountability tracking, consistent improvement was observed with academic engagement increasing to an average of 73% (range = 67–96%) and problem behavior decreasing to an average of 10% (range = 3–36%). Changes in variability of both behaviors were also observed. Upon implementation of CICO with accountability tracking, Evan displayed a reduction in variability of both target behaviors.

Jeremy initially showed moderate levels of appropriate academic engagement and problem behavior at baseline. Although data were relatively variable, there was an observable decreasing trend for academic engagement and increasing trend for problem behavior. Academic engagement occurred an average of 48% (range = 26–71%) of intervals, whereas problem behavior occurred an average of 33% (range = 5–63%) of intervals during baseline. Once the CICO intervention was introduced, a high degree of level change was observed in both academic engagement at an average of 62% (range = 44–84%) and problem behavior at an average of 19% (range = 1–42%). Starting day 14, Jeremy’s behaviors began to level out with little variability, close to the initial level observed during baseline. However, during this later phase of the intervention, Jeremy’s mean level of academic engagement (55%) was higher than that of baseline (48%), whereas the mean level of problem behavior (19%) was lower than that of baseline (33%). Unfortunately, after day 21, Jeremy was removed from the study due to moving to a new school.

James continued to operate at a relatively moderate level for both academic engagement (an average of 54%; range = 42–65%) and problem behavior (an average of 27%; range = 14–42%) during baseline. James displayed a noticeable level change immediately after he began participating in CICO. Although academic engagement showed a decreasing trend after a few days of implementation of the intervention, it consistently improved with an average of 74% (range = 43–97%) of intervals as days progressed and problem behavior decreased to 17% (range = 3–35%) of intervals. Changes in both behaviors were observed when a new coordinator conducted the morning and afternoon checks with the students in day 18; however, the behaviors returned to previous levels in day 19.

The results of Tau-*U* effect size estimates indicated that the magnitude of the intervention effect on student behaviors was small to large. Tau-*U* effect sizes ranged from 0.58 to 0.76 for academic engagement and 0.16 to 0.61 for problem behavior across students. More specifically, the Tau-*U* effect sizes for academic engagement were medium to large, with 0.65, 0.58, and 0.76 for Evan, Jeremy, and James, respectively, and with James showing the largest effect. The Tau-*U* effect sizes for problem behavior were 0.16, 0.56, and 0.61 for Evan, Jeremy, and James, respectively. Compared to academic engagement, the magnitude of the effect on problem behavior was small (Evan), medium (Jeremy), and large (James).

### 3.3. Student Behaviors during Fading

The number designation listed in the fading phase indicate the number of times the teacher provided feedback using the DRC throughout the day. The last fading phase for Evan was the time in which the coordinator was removed from the intervention and the student received only three instances of teacher feedback (morning, lunch, and prior to dismissal). When the fading was introduced, Evan’s academic engagement and problem behavior remained stable with an average of 68% (range = 63–72%) and 21% (range = 16–24%), respectively. Academic engagement and problem behavior remained stable during the second fading phase, with an average of 77% (range = 68–84%) for academic engagement and 18% (range = 9–27%) for problem behavior.

Upon the third fading phase, academic engagement further increased to 100% while problem behavior decreased to 0% at day 33. Although both behaviors did not remain at those levels in day 34, academic engagement remained at a higher level while problem behavior remained at a lower level during the following two days than those observed in the prior fading phase. For James, undesired behavioral changes were seen when attempting to move to the second fading phase. Behavior returned to appropriate levels after returning to the previous fading phase in which five segments were used. Overall, James maintained an average of 66% (range = 19–92%) of intervals with academic engagement and 14% (range = 0–36%) with problem behavior during the fading phases. Unfortunately, Jeremy was unable to continue to the fading phase as he discontinued the study due to his family moving out of the area.

### 3.4. Social Validity

At the conclusion of the study, participating students, parents, and teachers were given the student, parent, and teacher versions of the Behavior Education Program Acceptability Questionnaire to ascertain their opinions on the CICO intervention. Of the eight participants (three students, three parents, and two teachers), five participants (two students, one parent, and two teachers) completed the social validity questionnaires. Only Evan’s parent returned the social validity survey. The acceptability of the intervention procedures and satisfaction with the intervention outcomes were rated high, with an overall mean of 5.6 out of 6 (ranging from 4 to 6) across the participant groups. Overall scores indicated high acceptability and satisfaction across all participant groups (see Table 1), including willingness to participate in the intervention again, the likelihood of recommending the intervention to others, and successful changes in student behavior.

## 4. Discussion

This study evaluated the impact of the secondary tier intervention, CICO, on academic engagement and problem behavior for three elementary students from a high-need population, who were at-risk for developing severe problem behavior. The focus of the study was monitoring fidelity of implementing all components of the CICO intervention by coordinator, teachers, and parents to maximize the intervention outcomes and evaluating the maintenance effects while implementing fading procedures. The results indicated that all team members implemented the CICO with fidelity, except the initial phase of the intervention for Evan, and their implementation of intervention resulted in increasing academic engagement and reducing problem behavior for all three students. Tau-*U* Effect sizes were medium to large for academic engagement and small to medium for problem behavior across students. The accountability tracking implemented with one student (Evan) contributed to further improvement of target behaviors, and the intervention effects were maintained for two students who underwent fading. The CICO intervention also demonstrated a strong social validity. The students, teachers, and one parent all expressed satisfaction with the procedures and outcomes of the CICO intervention.

The findings were consistent with previous research in that school personnel could implement the CICO components with fidelity [9,17,18,25,39]. The implementation fidelity scores showed that all elements were fully in place by the coordinator and teachers. During implementation of the intervention, teachers anecdotally reported finding that the DRC served as a visual reminder for them to provide ongoing feedback to their students on their behavior progress throughout the day. This study also supports previous findings that CICO is effective in increasing academic engagement and reducing problem behavior of students in general education settings [9,17,19,26,40]. The study also contributes to the field by supporting at-risk urban elementary school students in a high-need school [41] where more than 75% of the students were receiving free or reduced-price lunch and approximately 91% of the students were from African-American or Hispanic families. All three participating students were from single-parent African-American families with incomes below the poverty line.

One important aspect of the present study is the maintenance effects demonstrated with two students. Removing the students from the DRC by systematically reducing the instances of teacher feedback throughout the day during fading were successful to maintain the improved behaviors. Furthermore, removing the coordinator was successful for one student (Evan). However, one student (James) did need to be returned to a previous fading phase, which could indicate that fading phases may need to be extended as changing levels of support could cause an adverse effect on target behaviors as shown in a component analysis by Campbell and Anderson [17]. Schools interested in implementing CICO should consider that some students may require being on CICO for an extended period of time. However, options to reduce resources (e.g., teacher time, use of daily reinforcers) for these at-risk students should be explored to sustain the intervention without risking staff burnout [40].

Although data are limited to one student, the present data suggest that for students who initially do not respond well to the implementation of CICO, modifying the intervention or additional supports may be needed to maximize treatment gains [9,42,43,44]. It was apparent that Evan’s academic engagement increased and problem behavior decreased with the initial implementation of CICO but not to the full extent possible. The use of student accountability tracking resulted in further improvement in his target behaviors. Compared to Jeremy and James, Evan had lower levels of academic engagement and higher levels of problem behaviors that were present along with a low rate of parent-teacher communication. By supplementing CICO with accountability tracking, it was possible to promote behavior change without requiring Tier 3 intervention supports. The accountability tracking contributed to increased Evan’s parent’s involvement, which may have led to increased his outcomes. Additional research is needed to further examine this finding.

Due to time constraints, the intervention could only be faded to three feedback sessions (without the coordinator) for one student, Evan. Jeremy was also unable to continue onto fading as his family moved to a new area, thus changing schools. For James, the intervention was only faded successfully to five feedback sessions. It is unknown if results would be maintained throughout a full removal of the intervention and at an additional follow-up time. Future research should look at systematically fading out the entirety of the intervention. Additionally, it should be noted that some difficulties were observed in the use of the DRC during fading of the CICO with accountability tracking. Evan reported to the CICO coordinator that punishment was being used at home by an extended family member with whom he had begun spending the afternoons. The family member was requiring that Evan meet a higher percentage goal on the DRC than what was set by the team at the school. When Evan made less than the family-expected goal, regardless of whether he made the goal set by the school team, punishments were delivered by the extended family member in the form of verbal reprimands, loss of privileges, and even corporal punishment. Attempts to work with the family to address these concerns were unsuccessful in maintaining progress confidentiality with only the mother. As a result, it was the decision of the school and the researcher to discontinue sending the DRC home for the final two fading phases. Instead, the coordinator would call home and discuss Evan’s progress with his mother at the end of the day. Evan reported that this was successful in alleviating the issue. Additional positive outcomes were seen for the students involved in the study. Throughout the intervention, students were observed encouraging one another to meet their point goals and at times were even effective in deescalating each other’s problem behaviors.

As discussed above, the major limitation of the current study is the limited demonstration of maintenance effects due to the small sample size and no examination of a long-term maintenance effect. With time constraints and the state-wide annual student assessment, a small number of students were recruited to participate in the study, and data collection opportunities were often restricted. More research is needed to evaluate the external validity of the current findings. Another limitation is that there was only 1 day overlap between Jeremy’s baseline and Evan’s intervention and only 2 days overlap between Jeremy’s baseline and Evan’s intervention. Additional baseline data should have been collected for both students until previous students receiving intervention demonstrate clear improvement in behavior. On the day when the intervention was introduced to Jeremy, Evan’s academic engagement decreased and problem behavior increased close to baseline mean levels. To demonstrate strong experimental control, the treat to internal validity such as maturation and coincidental events should have been controlled before introducing intervention to the next tiers with respect to participants, having sufficient time lag between phases across tiers [45]. Given the recent What Works Clearinghouse (WWC) standards for single case design studies [46], efforts should be made in future studies to address the design standards and evidence criteria in the standard.

Despite its limitations, this study offers a contribution to the body of research on CICO. This study is one of the few studies that examined the maintenance effects of CICO by systematically fading the number of checks in using DRC by teachers. This study is also one of the few studies that promoted parent involvement by assessing fidelity of using DRC by parents to enhance the effects of CICO [19,20,25]. Although CICO is an effective Tier 2 intervention that is applicable with students from high-need backgrounds, supplementing the intervention with additional strategies similar to the student accountability tracking used in this study could be a promising option for the Tier 2 intervention. Increasing the parent use of the DRC through student accountability tracking may be a viable tool for promoting home–school collaboration and enhancing behavioral outcomes for students who are not responding to the standard CICO.

## Figures and Tables

**Figure 1 behavsci-13-00257-f001:**
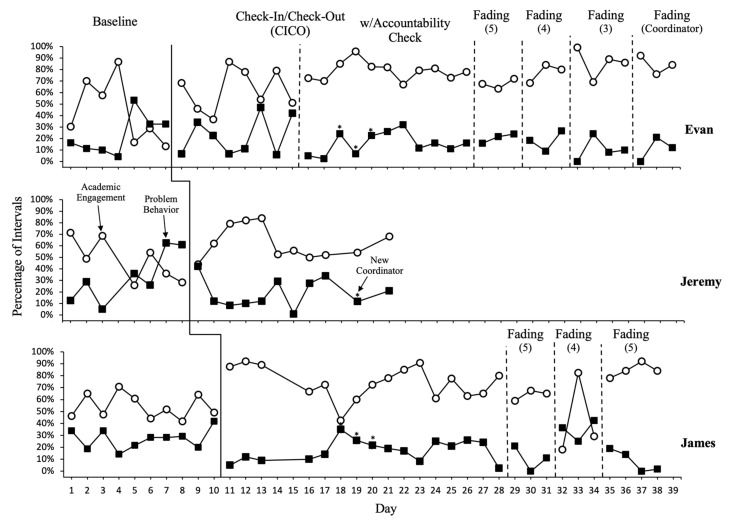
Percentage of intervals with academic engagement and problem behavior across conditions and students. The asterisks refer to the days when a new coordinator implemented CICO.

**Table 1 behavsci-13-00257-t001:** Summary of social validity data by each group.

	Student *M* (Range)	Parent *M* (Range)	Teacher *M* (Range)
Acceptability of CICO Procedures			
Ease of implementation Usefulness of the intervention	6 (6–6) 6 (6–6)	6 (6–6) 6 (6–6)	5.5 (5–6) 5.5 (5–6)
Satisfaction with Outcomes			
Increases in academic engagement Decreases in problem behavior	5 (4–6) 6 (6–6)	6 (6–6) 6 (6–6)	4.5 (4–5) 4.5 (4–5)
Overall	5.8	5.9	5.1

## Data Availability

Data are available on reasonable request from the corresponding author.

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
