# Peer review of "Implementation of Check-In/Check-Out to Improve Classroom Behavior of At-Risk Elementary School Students"

_behavsci, 2023, doi:10.3390/bs13030257_

Round 1
Reviewer 1 Report
Thank you for the opportunity to review "Implementation of Check-In/Check-Out to Improve Classroom Behavior...". The manuscript is well written with three purposes of study: fidelity of implementation; student increase in engagement and decrease in behavior events; maintenance of behavior changes during fading. There are some fatal flaws within the methodology.
First, effect size for the intervention is not calculated or reported. The range for each student is large and variable. By reporting some measure of effect size (See Parker, Vannest, & Davis, 2011; Lenz (2012 is also helpful though that was in counseling), the reader could be sure that the intervention was indeed effective. Otherwise it is impossible to objectively understand the effect of the treatment and the answer to questions 2 and 3.
Second, there appears to be assumptions that do not seem possible by the information that is provided. For example, it is impossible to know if the parent reviewed the DRC and provided feedback by the use of a signed DRC. (page 6 line 299, page 7 line 300). Jeremy didn't get fading so the statement starting line 523 doesn't match the data.
Third, the measure for social validity comes from a reference that is a meta-analysis. The validity information for such a measure should be included. Are there three versions (student, parents, teachers) within that reference? In reviewing that reference, it is unclear how it is used for the Behavior Education Program Acceptability Questionnaire.
Fourth, there is not enough information provided about the methodology. How many days of baseline was necessary and at what level prior to intervention (when did intervention start for students)? What was used to determine which student would start first? Why does it appear that Evan and Jeremey started intervention at the same point within a concurrent multiple baseline design? How was fidelity from the CICO and teacher coordinator collected? Did observers watch them every day of student intervention throughout the data as well? What were the goals for each student? What were the fidelity goals for teachers and parents? Why isn't the replacement guidance counselor included in the description of participants? As there are potential threats to the use of a concurrent baseline, addressing how the methodology and design was implemented to avoid these threats is essential. See Slocum et al. 2022.
Fifth, it would be helpful to have data for question 1 and social validity data in a table. The data by type of participant for social validity would be helpful. Additionally the figure is hard to read as it is small so differences by percentage is hard to discern. An additional line for Evan with the addition of the accountability check would be helpful.
There are some minor considerations for the authors to address.
1. line 128 page 3 should be (c) for the third research question.
2. Line 234 page 5 should be 30s.
3. All parents were single moms. Why is dad included line 415 page 9? And parents line 148?
4. There are four steps described starting line 409 page 9 but the list is a, b, d, e.
5. The figure should have a key and not be embedded in Jeremy's graph.
Reviewer 2 Report
I greatly appreciate having the opportunity to review the manuscript, “Implementation of Check-In/Check-Out to Improve Classroom Behavior of At-Risk Elementary School Students.” The authors investigated the effects of a Tier 2 intervention, Check-In/Check-Out (CICO), on three elementary school students. As a researcher who was an educator, I was very excited to review the manuscript, and I appreciated the authors’ effort to focus on implementation fidelity and maintenance of this intervention. I believe the authors systematically investigated the intervention in an applied setting with a valid single-case design, and the findings are meaningful to the behavior analytic and education community. Please find the questions and suggestions below.
Introduction
1. Brief descriptions of Tier 1 and Tier 3 interventions may help novel readers understand PBIS better. This would also be helpful because the Discussion section mentions Tier 3 intervention supports [line 583].
2. While I appreciate the thorough description of Tier 2 interventions, adding a couple of Tier 2 intervention examples may be helpful to the readers new to PBIS (e.g., small group instructions, social skills group).
3. Overall, the introduction is well-written and clear. The authors did a nice job at including the relevant literature.
Method
1. [page 5, line 234] Please edit “30th” to “30s”
2. [page 7, line 333] Please edit “ranting” to “ranging”
3. [page 9, lines 403-407] Further clarification of the description of minimal changes in Evan’s target behavior would be beneficial. This part sounds like the paragraph started to report the results rather than describing the intervention procedure. Perhaps, rephrasing the sentences, so the modifications that were made come first may help clarify. For example, “accountability-tracking system was introduced to Evan’s intervention phase because of…. (see results for further detail)”
Results
1. [page 10, line 495] I suggest avoiding the use of the word “significant” in the results section unless there was a statistical significance found.
Discussion & Conclusion
1. Another limitation that should be noted is the shorter verification period across participants. There was only one day overlap between Jeremy’s baseline and Evan’s intervention and only two days overlap between James’ baseline and Jeremy’s intervention. I do not think it is a limitation that affects the overall quality of the study since all students showed improvement right after CICO was introduced. However, discussing it from an experimental control standpoint may be worth it.
Tables & Figures
1. The phase change lines are misleading as it seems like Jeremy and James both went through Fading (5), (4), and (3) at the same time as Evan did. I suggest removing the phase change line from Jeremy’s panel, and only keeping one phase change line for James when Fading (5) begins.
Again, I appreciate the opportunity to review this manuscript and I sincerely hope the authors will find my review helpful.
Round 2
Reviewer 1 Report
Thank you for your responses and your edits to the manuscript which strengthens the quality.